# CONSISTENT123: IMPROVE CONSISTENCY FOR ONE IMAGE TO 3D OBJECT SYNTHESIS

## ABSTRACT

Large image diffusion models enable novel view synthesis with high quality and excellent zero-shot capability. However, such models based on image-to-image translation have no guarantee of view consistency, limiting the performance for downstream tasks like 3D reconstruction and image-to-3D generation. To empower consistency, we propose *Consistent123* to synthesize novel views simultaneously by incorporating additional cross-view attention layers and the shared self-attention mechanism. The proposed attention mechanism improves the interaction across all synthesized views, as well as the alignment between the condition view and novel views. In the sampling stage, such architecture supports simultaneously generating an arbitrary number of views while training at a fixed length. We also introduce a progressive classifier-free guidance strategy to achieve the trade-off between texture and geometry for synthesized object views. Qualitative and quantitative experiments show that Consistent123 outperforms baselines in view consistency by a large margin. Furthermore, we demonstrate a significant improvement of Consistent123 on varying downstream tasks, showing its great potential in the 3D generation field.

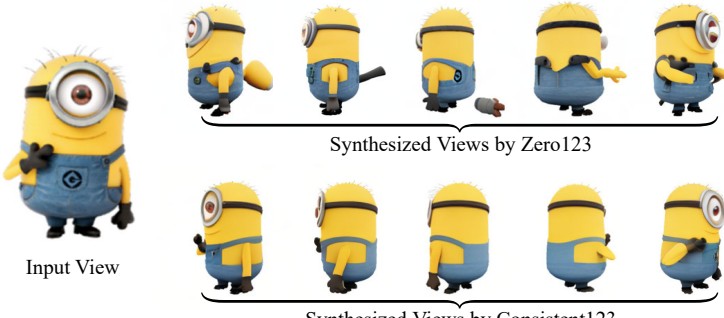

Figure 1: Given the input view and relative pose sequence, Consistent123 can synthesize consistent novel views concurrently, while Zero123 fails at producing consistent views.

## 1 INTRODUCTION

Large diffusion models (Ramesh et al., 2022; Saharia et al., 2022; Rombach et al., 2022; Balaji et al., 2022) have achieved remarkable performance in synthesizing high-quality images with significant zero-shot capability. To migrate such generalizability to the 3D field, diffusion models have been adapted to 3D object synthesis (Watson et al., 2023; Zhou & Tulsiani, 2023; Tseng et al., 2023; Liu et al., 2023b; Xiang et al., 2023). To utilize the capability of diffusion architecture and the pre-trained weights on large-scale images, the 3D synthesis process of these methods is naturally formulated as the image-to-image translation from input view to target view with pose condition. Such formulation yields excellent image quality but lacks alignment among synthesized views, leading to the loss of view consistency.

Some methods (Watson et al., 2023; Tseng et al., 2023) attempt to improve the view consistency by auto-regressively generating novel views, *i.e.*, conditioning on previously generated views during

view synthesis. While there has been some improvement in consistency, these methods still suffer from the issues of accumulated errors and slow inference speed, struggling in the loop closure. Furthermore, most of these methods are trained on a single or limited object categories (like ShapeNet), thus failing at generalization to all kinds of object classes. *Therefore, how to generate consistent multi-views for objects of arbitrary categories still remains unexplored.*

In this paper, we propose *Consistent123*, which generates multiple novel views simultaneously given the condition view and a sequence of relative pose transformations. We carefully design the attention mechanism inside the denoising U-Net. First, we introduce additional cross-view attention layers as Video Diffusion Model (Ho et al., 2022) to allow interaction across all synthesized views. This allows the model to synthesize views corresponding to an underlying 3D object, thus significantly improving the consistency. Moreover, previous works like Zero123 (Liu et al., 2023b) are conditioned on the semantic embedding (i.e., CLIP embedding), omitting the spatial layout information of the condition view. To better align with the condition view, we introduce shared self-attention inspired by (Cao et al., 2023), which shares the key and value in the self-attention layers of the condition view with synthesized novel views. Note that the shared self-attention mechanism is applied in both the training and inference stages without introducing additional trainable parameters. For efficient training, the spatial layers of our model are initialized with the Zero123 pre-trained weights and only the cross-view attention weights are trainable to preserve the zero-shot capability.

At the sampling stage, Consistent123 synthesizes all the novel views concurrently given the condition view and relative poses. Since synthesized views are aligned only via attention layers, our model can sample arbitrary numbers of views while training at a fixed length. Furthermore, we found surprisingly that denoising a large number of views simultaneously is crucial for view consistency. With more sampling views, there is more overlapping and interaction across views thus the consistency is better. This contradicts the auto-regressive methods (Watson et al., 2023), whose sampling quality usually worsens with increasingly more conditioning views. Besides the arbitrary sampling, we also observe that the scale of classifier-free guidance (CFG) (Ho & Salimans, 2022) greatly influences the sampling quality. Specifically, a large CFG scale contributes to the geometry of synthesized object views while a low CFG scale helps to refine the texture details. Therefore, we propose progressive classifier-free guidance (PCFG), dynamically decreasing the guidance scale during the denoising process to achieve the trade-off between geometry and texture. With these sampling strategies, the consistency and quality of synthesized novel views are further boosted.

The proposed model is evaluated on multiple benchmarks, including Objaverse (Deitke et al., 2023) testset, GSO (Downs et al., 2022), and RTMV (Tremblay et al., 2022). We evaluate both the view consistency score (Watson et al., 2023) and the similarity with ground truth on these datasets. Furthermore, we demonstrate performance improvement in downstream applications like 3D reconstruction and image-to-3D generation. We hope that Consistent123 will be a foundational model for further 3D generation research.

The contribution of our paper can be summarized as follows:

- We propose Consistent123, which synthesizes consistent novel views simultaneously conditioned on the input view and relative poses. We introduce cross-view attention and shared self-attention mechanisms to achieve view alignment for better consistency.
- We show that Consistent123 supports synthesizing an arbitrary number of views. We also propose progressive classifier-free guidance to achieve the trade-off between geometry and texture.
- We demonstrate a great improvement aided by Consistent123 on multiple downstream tasks like 3D reconstruction and image-to-3D generation.

## 2    RELATED WORK

**Geometric-aware 3D Object Synthesis.**    The primary solution of 3D object synthesis is to generate 3D representation directly, including NeRF-class methods (Yu et al., 2021; Niemeyer et al., 2022; Jang & Agapito, 2021), tri-planes (Gao et al., 2022; Skorokhodov et al., 2023; Sargent et al., 2023) and point clouds (Luo & Hu, 2021; Nichol et al., 2022; Zeng et al., 2022). With these geometric-aware models, the objects are synthesized via volume rendering, and the view consistency is guaranteed by construction. While the consistency is preserved, most works are designed for some well-aligned specific datasets, thus difficult to generalize for various objects. Only a tiny portion of

methods (Skorokhodov et al., 2023; Sargent et al., 2023) can achieve multi-classes object synthesis, but their performance is constrained by model capacity and 3D data scale, leading to low image quality and limited generalizability.

**Geometric-free 3D Object Synthesis.** Geometry-free methods formulate the 3D object synthesis as the image-to-image translation. Early methods (Rombach et al., 2021; Sajjadi et al., 2022; Kulhánek et al., 2022) are mainly based on Transformer (Vaswani et al., 2017) architecture. With the fast development of diffusion models, they are shown to have a high capability to synthesize high-quality images (Ho et al., 2020; Nichol & Dhariwal, 2021; Dhariwal & Nichol, 2021; Song et al., 2022; Karras et al., 2022). Recent works (Watson et al., 2023; Zhou & Tulsiani, 2023; Tseng et al., 2023; Liu et al., 2023b) try to achieve novel view synthesis with the aid of diffusion models. Zero123 (Liu et al., 2023b) finetunes pre-trained diffusion model on the large-scale 3D dataset to support the camera control. For view consistency, Watson et al. (2023) proposed stochastic conditioning to make the synthesized view aligned with previous views auto-regressively, while Zhou & Tulsiani (2023) introduced additional distillation loss to optimize the final 3D representation. With these post-processing methods, the view consistency of models is improved at the expense of sampling flexibility, limiting the downstream applications of these models.

Unlike the above-mentioned methods, the proposed Consistent123 synthesizes all consistent views simultaneously corresponding to an underlying object. Very recently, the concurrent work MVDiffusion (Tang et al., 2023) shares a similar spirit to synthesize all views simultaneously. However, it is used for the text-to-scene task while our model is designed for generic 3D object synthesis, and the attention mechanism inside the model significantly differs.

# 3 METHODS

This section first discusses the formulation of previous pose-guided image-to-image diffusion models in Section 3.1. Next, we present Consistent123 and the proposed attention mechanism in Section 3.2. Finally, two sampling techniques are introduced in Section 3.3 to boost image consistency and quality further.

## 3.1 POSE-GUIDED IMAGE-TO-IMAGE DIFFUSION MODEL

Diffusion models (Ho et al., 2020) are probabilistic generative models that recover images from a specified degradation process. The forward process adds Gaussian noise to the target image $x$:

$$q(x^t|x^{t-1}) = \mathcal{N}(x^t; \sqrt{\alpha_t}x^{t-1}, (1 - \alpha_t)I) \tag{1}$$

Here, $\alpha$ is a scheduling hyper-parameter and the diffusion timestep $t \in [1, 1000]$. Given the condition image $x_c$ and the relative pose transformation from the condition view to the target view $\Delta p = (R, T)$, the denoising process can be defined as follows:

$$p(x^{t-1}|x^t, c(x_c, \Delta p)) = \mathcal{N}(x^{t-1}; \mu_\theta(x^t, t, c(x_c, \Delta p), \Sigma_\theta(x^t, t, c(x_c, \Delta p)))) \tag{2}$$

where $c(x_c, \Delta p)$ represents the embedding of the condition image and relative pose, the mean function $\mu_\theta$ is modeled by the denoising U-Net $\epsilon_\theta$ and the variance can be either predefined constants or trainable parameters. The training target is to minimize the reconstruction loss of noise $\epsilon$ sampled from the standard normal distribution:

$$L(\theta) = \min_\theta \mathbb{E}_{x,t,\epsilon \sim \mathcal{N}(0,1)} \|\epsilon - \epsilon_\theta(x^t, t, c(x_c, \Delta p))\|_2^2 \tag{3}$$

**Challenges.** Leveraging the pre-trained image diffusion models can partially alleviate the generalization of 3D objects caused by the limited 3D data scale and model capability. However, such geometry-free models typically suffer from the alignment problem that the consistency across synthesized views can not be guaranteed. Even if some auto-regressive sampling techniques can improve the view consistency, they heavily limit the sampling process and further applications. It remains unexplored how to synthesize 3D objects with zero-shot capabilities and view consistency.

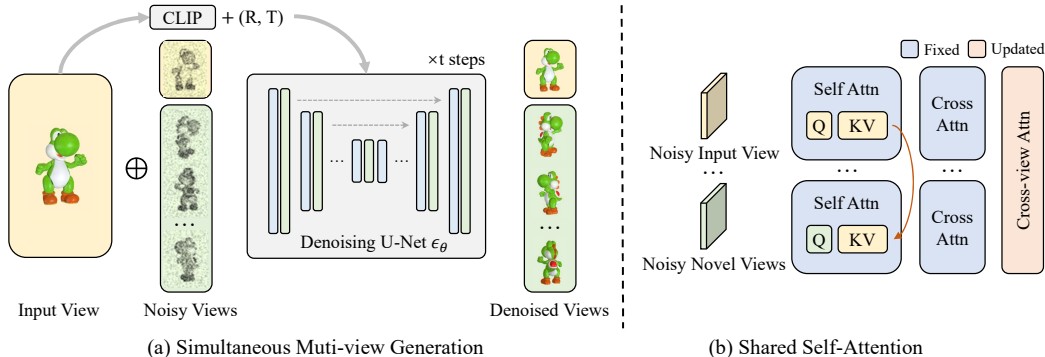

(a) Simultaneous Muti-view Generation                    (b) Shared Self-Attention

Figure 2: **The overall method of Consistent123.** (a) At the training stage, multiple noisy views concatenated (denoted as ⊕) with the input view are fed into the denoising U-Net simultaneously, conditioned on the CLIP embedding of the input view and the corresponding poses. For sampling, views are denoised iteratively from the normal distribution through the U-Net. (b) In the shared self-attention layer, all views query the same key and value from the input view, which provides detailed spatial layout information for novel view synthesis. The input view and related poses are injected into the model by the cross-attention layer, and synthesized views are further aligned via the cross-view attention layer.

## 3.2 VIEW CONSISTENCY VIA LATENT ALIGNMENT

**Simultaneous Multi-view Generation.** Previous works show the excellent quality of image-based diffusion models in synthesizing novel views. However, most of them lack view consistency due to the image-to-image translation formulation, preventing the view interaction by design. We instead propose to synthesize a sequence of novel views simultaneously to facilitate the interaction of different novel views, which greatly improves the consistency among the generated novel views. Given a sequence of target novel views $X = (x_1, x_2, ..., x_n)$ and relative pose sequence $\Delta P = (\Delta p_1, \Delta p_2, ..., \Delta p_n)$, the denoising process can be adapted from Equation 2 as:

$$p(X^{t-1}|X^t, c(x_c, \Delta P)) = \mathcal{N}(X^{t-1}; \mu_\theta(X^t, t, c(x_c, \Delta P)), \Sigma_\theta(X^t, t, c(x_c, \Delta P))) \qquad (4)$$

Here $n$ is the number of views used for simultaneous view generation. Sepeically, we use Zero123 (Liu et al., 2023b) as our base model and incorporate an additional trainable cross-view attention layer after each self-attention layer within the denoising U-Net to uphold the learning of view consistency. The weights of other components are simply loaded from the pre-trained Zero123 and kept fixed during training.

**Shared Self-attention.** Zero123 (Liu et al., 2023b) employs two ways for utilizing conditioning image: one involves concatenating it with input noise, and the other is applying the cross-attention with its CLIP image embedding. For the first way, (Watson et al., 2023) argues that it is insufficient in enhancing view consistency, especially when there exists a large view difference. The second way provides the high-level semantic information of the conditioning view for the generator while ignoring its detailed spatial layout information. Inspired by (Cao et al., 2023; Wu et al., 2023), we propose to enable each target view to attend to the input view when performing self-attention. More specifically, each target view queries the key and value of the input view for all self-attention layers of denoising U-Net. Assume that $Q, K, V$ are the query, key, and value matrices projected by the spatial feature of view latent. The $i^{th}$ novel view only calculates its query matrix $Q_i$ to attend with the shared key and value matrices of condition image $K_c, V_c$. The self-attention result $A_i$ of $i^{th}$ novel view can be formulated as follow:

$$A_i = \text{softmax}(\frac{Q_i K_c^T}{\sqrt{d}})V_c \qquad (5)$$

This operation aligns the synthesized views with the input image without requiring additional trainable parameters, thus preventing further overfitting. Besides, it reduces part of the computation of self-attention layers by sharing condition intermediates $K_c$ and $V_c$. For implementation, we feed the condition image and $n-1$ novel views into the U-Net. See Figure 2(b) for a visual depiction.

### 3.3 Sampling Techniques for Novel View Synthesis

**Arbitrary-length Sampling.**  Recall that in Section 3.2, a fixed number of views is fed into the denoising U-Net simultaneously for training. In the sampling stage, it is natural to sample the same number of views as training. But is it possible to synthesize more consistent views at once? Since our model connects different views only via cross-view attention layers, this design allows us to sample views in arbitrary lengths without further modification. We surprisingly find that if the number of sampling views is expanded, the consistency of views also significantly *increase*. This contradicts the auto-regressive methods, whose sampling quality usually becomes *worse* with the increasing conditioning views (Watson et al., 2023). It is expected since the cross-view attention works better with closer view intervals (i.e., smaller view differences). With this fantastic feature, our model can synthesize more than 64 consistent views simultaneously (even trained at 8 views), which satisfies most of the application scenarios. As for an extremely large number of views (e.g., over 256) or low-memory devices, a compromise solution is to first synthesize an acceptable number of views, and then predict their nearby views for the next round.

**Progressive Classifier-free Guidance.**  Classifier-free Guidance (Ho & Salimans, 2022) is a commonly used sampling technique for diffusion models.

$$\tilde{\epsilon}_\theta(z_t, c(x_c, R, T)) = (1 + w)\epsilon_\theta(z_t, c(x_c, R, T)) - w\epsilon_\theta(z_t) \tag{6}$$

Here $w$ is a scalar to control the used guidance scale. As shown in Figure 3, we found that a small $w$ (e.g., 1.5) helps to synthesize detailed textures but with unacceptable artifacts. On the contrary, a large $w$ (e.g., 10) can guarantee excellent object geometry at the expense of textures. Based on this observation, we propose Progressive Classifier-free Guidance (PCFG) schedule to decrease $w$ during the denoising process. The basic intuition is that the model synthesizes the object outlines at the early denoising stage, where a large $w$ can promise the geometry and reduce the artifacts. At the later denoising stage, the model concentrates on refining the object details thus a

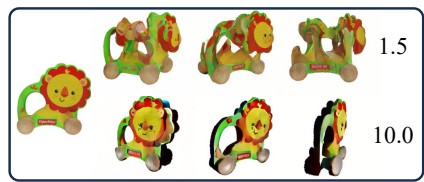

Figure 3: **The synthesized views under different CFG scales.**

relatively small $w$ can improve the synthesized textures. We experiment with varying types of reduction functions, including linear, concave, and convex functions:

$$w(t) = \begin{cases} (w_e - w_s)\frac{t}{T} + w_s & \text{(linear)} \\ (w_e - w_s)\frac{t^2}{T^2} + w_s & \text{(concave)} \\ w_s\frac{1}{t} + w_e & \text{(convex)} \end{cases} \tag{7}$$

where $T$ represents the total denoising steps (commonly 50 steps), $w_s$ is the starting CFG scale and $w_e$ is the ending CFG scale. We found empirically that the concave function performs best.

## 4 Experiment Results

### 4.1 Experiment Setup

**Benchmarks.**  Following Zero123 (Liu et al., 2023b), we use Objaverse (Deitke et al., 2023) to finetune our model, which is a large-scale open-source dataset containing 800K+ 3D models created by 100K+ artists. Besides, for more efficient learning of the consistency among views, we circularly render 18 views for each object with $15°$ perturbation in elevation (i.e., the elevation angles range in $[75°, 105°]$). The azimuth angle interval between the nearest views is randomly set at $[10°, 30°]$. We separate $1\%$ of the rendering for testing. Follow the settings of Liu et al. (2023b), the performance of our model is evaluated on multiple benchmarks, including Objaverse (Deitke et al., 2023) testing set, GSO (Downs et al., 2022) and RTMV (Tremblay et al., 2022). We randomly picked up 100 objects from Objaverse testset, and 20 from both GSO and RTMV as Zero123.

**Metrics.**  Considering that our main contribution is to improve the view consistency, we use the *3D consistency score* (Watson et al., 2023) to evaluate the model performance. Specifically, views are first sampled from the models given the condition image and relative poses, then a NeRF-like

| Dataset | Objaverse Testset | | | GSO | | | RTMV | | |
|---|---|---|---|---|---|---|---|---|---|
| Model | PSNR↑ | SSIM↑ | LPIPS↓ | PSNR↑ | SSIM↑ | LPIPS↓ | PSNR↑ | SSIM↑ | LPIPS↓ |
| Zero123 | 21.72 | 0.92 | 0.23 | 22.88 | 0.92 | 0.25 | 15.68 | 0.78 | 0.36 |
| Zero123 + SC | 22.09 | 0.92 | 0.21 | 22.30 | 0.93 | 0.21 | 15.88 | 0.76 | 0.36 |
| Consistent123 | **24.98** | **0.96** | **0.14** | **27.98** | **0.98** | **0.11** | **18.76** | **0.85** | **0.25** |

Table 1: **The overall comparison on consistency score.** The proposed Consistent123 significantly improves the view consistency compared with baselines by a large margin. SC means stochastic conditioning.

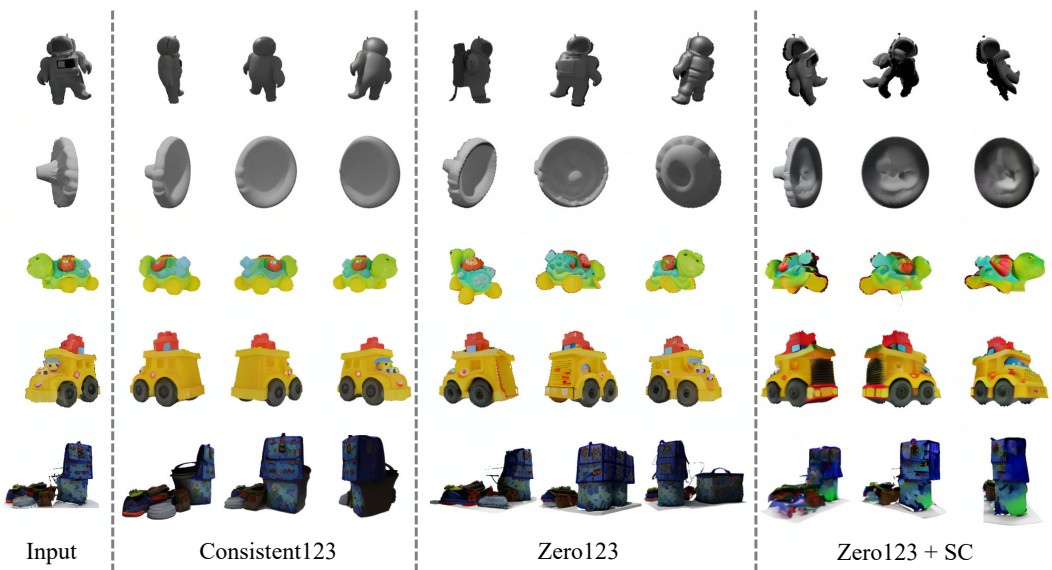

Figure 4: **Qualitative comparison with baselines from different datasets.** Two objects from Objaverse testset, two are from GSO, and the final one is from RTMV. Consistent123 synthesizes consistent multi-views while preserving high image quality.

neural field is trained on parts of the sampling views and evaluated on the remaining views. The higher performance the trained NeRF achieves, the more consistent the synthesized views are with each other. Here we use Instant-NGP (Müller et al., 2022) as the NeRF implementation to accelerate the metric calculation. We use the following metrics to evaluate the performance of methods quantitatively: PSNR, SSIM (Wang et al., 2004), LPIPS (Zhang et al., 2018). Besides, for more comprehensive experiments, we also compare the sampling views with the ground truth using the above metrics.

**Baselines.** We compare our model with Zero123 (Liu et al., 2023b), a strong geometric-free baseline in zero-shot 3D object synthesis with released pre-trained weight on Objaverse. Since 3DiM (Watson et al., 2023) has not been released, we only apply stochastic conditioning (SC) on Zero123 to compare with its improvement in consistency.

**Implementation Details.** For efficient training, we initialize the spatial weight of Consistent123 with Zero123 pre-trained weights. The cross-view attention layers are additionally incorporated into Consistent123 and trained on the renderings of Objaverse. We use AdamW (Loshchilov & Hutter, 2017) as the optimizer with $\beta_1 = 0.9$ and $\beta_2 = 0.999$ with a learning rate of $10^{-4}$. Follow Liu et al. (2023b), we reduce the image size to $256 \times 256$ and the corresponding latent dimension to $32 \times 32$. The total sampling step $T$ for all experiments is set to 50 with DDIM (Song et al., 2022) sampler. We trained our model on an 8×A100-80GB machine for around 1 day.

| Dataset | Objaverse Testset | | | GSO | | | RTMV | | |
|---|---|---|---|---|---|---|---|---|---|
| Model | PSNR↑ | SSIM↑ | LPIPS↓ | PSNR↑ | SSIM↑ | LPIPS↓ | PSNR↑ | SSIM↑ | LPIPS↓ |
| Consistent123 | **24.98** | **0.96** | **0.14** | **27.98** | **0.98** | **0.11** | **18.43** | **0.85** | **0.25** |
| - Cross-view Attn | 21.94 | 0.92 | 0.22 | 23.04 | 0.94 | 0.22 | 15.22 | 0.77 | 0.35 |
| - Shared Self-attn | 22.23 | 0.92 | 0.20 | 24.64 | 0.94 | 0.17 | 17.14 | 0.81 | 0.31 |
| - PCFG | 24.38 | 0.95 | 0.15 | 27.20 | 0.97 | 0.12 | 18.02 | 0.84 | 0.26 |

Table 2: **Overall ablation study on consistency score.** One of three components (cross-view attention, shared self-attention, and progressive classifier-free guidance) is removed for each experiment.

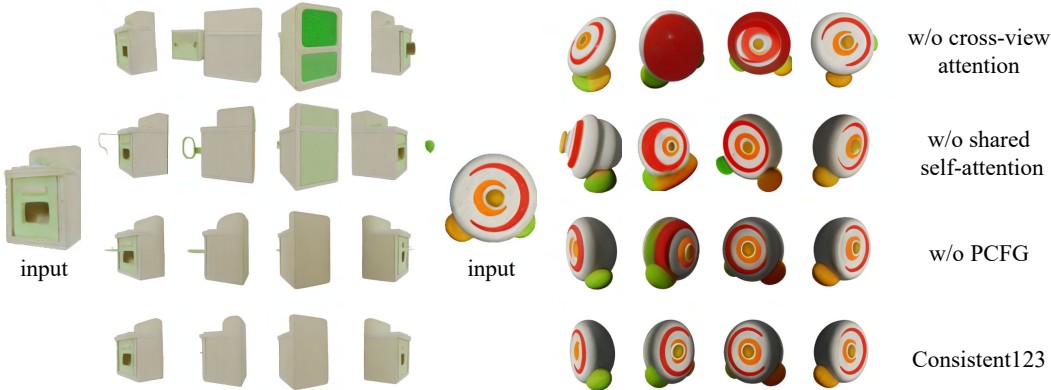

Figure 5: **Qualitative ablation study for different components.**

## 4.2 CONSISTENCY EVALUATION

We first evaluate the 3D consistency score of the Consistent123 model and several baselines. To promise constructed NeRF quality, we sample 64 views for each object, where 7/8 views are used for NeRF training and others for metrics evaluation. We use Objaverse Testset to evaluate the in-distribution performance and GSO and RTMV to evaluate the out-of-distribution performance. Zero123 (Liu et al., 2023b) is used as a strong baseline for novel view synthesis. And we apply stochastic conditioning (SC) sampling (Watson et al., 2023) on Zero123 auto-regressively to improve the view consistency further. As shown in Table 1 and Figure 4, the proposed Consistent123 significantly improves the view consistency compared with baselines by a large margin for either in-distribution or out-of-distribution tests. From these results, Zero123 synthesizes high-quality but inconsistent novel views. Stochastic conditioning helps to improve the view consistency but at the expense of image quality. By building the connection among synthesized views, the proposed Consistent123 can synthesize consistent multi-views without losing the image quality.

## 4.3 ABLATION STUDY

**Overall Ablation Study.** The ablation study is conducted for three important components of Consistent123: cross-view attention, shared self-attention, and progressive classifier-free guidance. For each ablation experiment, the consistency score is evaluated on several benchmarks, with one of these three components removed and the others unchanged. As shown in Table 2 and Figure 5, the cross-view attention is the most important component in Consistent123, revealing that interaction across all views is crucial to promise consistency. The shared self-attention layers further connect the condition image with synthesized images, enhancing the alignment of view content. Progressive classifier-free guidance also helps to eliminate the artifacts of synthesized view while preserving image quality, which slightly improves the model consistency.

**The Number of Sampling Views.** Aided with arbitrary-length sampling in Section 3.3, we can sample views in arbitrary length while training at the fixed number of views. As shown in Figure 6, we surprisingly found that as the number of sampling views increased, the synthesized views were

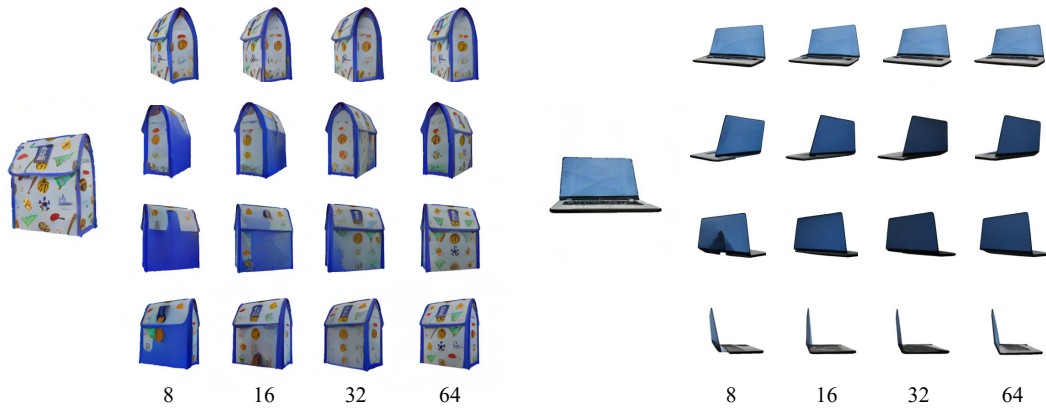

Figure 6: **The consistency comparison with different numbers of sampling views.** As the number of sampling views increases, the view consistency and quality are also improved due to the higher overlapping and interaction across views.

| Type | PSNR↑ | SSIM↑ | LPIPS↓ |
|------|-------|-------|--------|
| Linear | 27.05 | 0.97 | 0.12 |
| Convex | 24.07 | 0.94 | 0.14 |
| Concave | **27.98** | **0.98** | **0.11** |

(a) Types of reduction functions in PCFG.

| Layers | PSNR↑ | SSIM↑ | LPIPS↓ |
|--------|-------|-------|--------|
| Encoder | 25.92 | 0.96 | 0.14 |
| Decoder | **27.98** | **0.98** | **0.11** |
| Whole U-Net | 27.60 | **0.98** | **0.11** |

(b) U-Net layers for shared self-attention.

Table 3: **Ablation study of components on GSO.** (a) The consistency score of different types of reduction functions in PCFG. The concave reduction function can achieve the best performance among the candidates. (b) The consistency score of different U-Net layers that apply shared self-attention. Only applying shared self-attention in U-Net decoder layers can achieve the best performance.

more consistent. It is expected since there is more overlapping and interaction across views as the number of sampling views increases.

**Types of reduction functions in PCFG.** For the reduction functions in Equation 7, we empirically set $w_s = 10$ and $w_e = 2$. Table 3(a) shows the performance of different types of reduction functions, including linear, convex, and concave functions. The concave function achieves the best performance, showing that building the layout quickly and refining the details with more steps is a better strategy to improve the view consistency.

**U-Net layers for shared self-attention.** In Table 3(b), we analyze different layers in denoising U-Net for shared self-attention. Applying shared self-attention only in the U-Net decoder is slightly better. The main reason is that the query features in shallow layers of U-Net (e.g., encoder part) may not contain clear spatial layout information, thus the synthesized views are dominantly influenced by the conditioned view.

## 4.4 NOVEL VIEW SYNTHESIS EVALUATION

Besides view consistency, the quality of each synthesized view is also important for 3D objects. As common settings, we calculate the similarity metrics between novel views with ground truth. Similar to previous sections, we synthesize 64 views for each object and compare them with the object renderings. Note that the number of our evaluation views slightly differs from Liu et al. (2023b), leading to a small difference in the final metrics. In Table 4, Consistent123 outperforms most of the baselines in novel view synthesis. This demonstrates that Consistent123 can improve view consistency without losing the quality of synthesized views.

| Dataset | Objaverse Testset | | | GSO | | | RTMV | | |
|---|---|---|---|---|---|---|---|---|---|
| Model | PSNR↑ | SSIM↑ | LPIPS↓ | PSNR↑ | SSIM↑ | LPIPS↓ | PSNR↑ | SSIM↑ | LPIPS↓ |
| Zero123 | 15.52 | 0.85 | 0.15 | 17.42 | 0.86 | 0.15 | 10.10 | 0.64 | **0.33** |
| Zero123 + SC | 14.90 | 0.84 | 0.18 | 16.37 | 0.85 | 0.18 | **10.44** | 0.64 | 0.33 |
| Consistent123 | **16.46** | **0.86** | **0.14** | **18.22** | **0.87** | **0.13** | 9.87 | **0.66** | 0.34 |

Table 4: **The overall comparison on ground truth renderings.** Note that the number of our evaluation views slightly differs from Zero123, leading to a small difference in the final metrics.

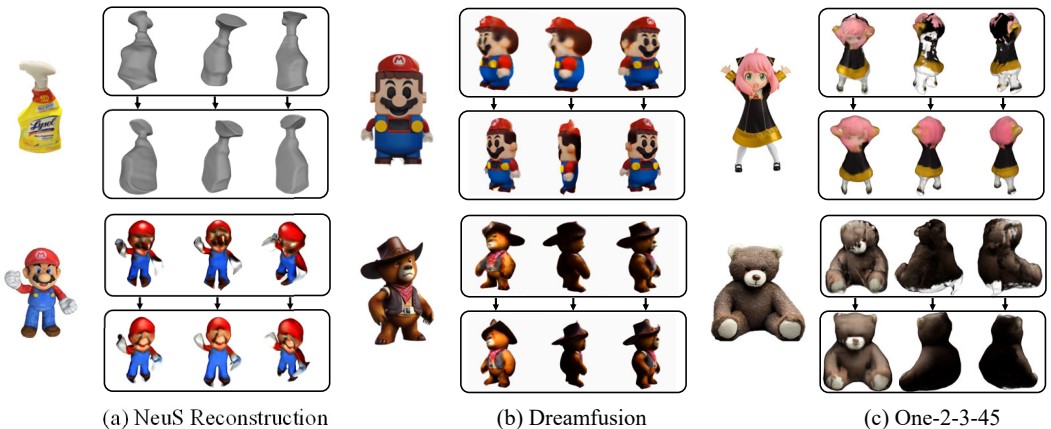

(a) NeuS Reconstruction  (b) Dreamfusion  (c) One-2-3-45

Figure 7: **Boosted 3D lifting results with Consistent123.** We experiment with varying downstream tasks, including NeuS (Wang et al., 2021), Dreamfusion (Poole et al., 2023), and One-2-3-45 (Liu et al., 2023a). For each object, the first row is from Zero123 while the second row is from Consistent123 (the texture of the first Neus case is removed to show the geometry quality, which can be found in the supplementary). The 3D lifting performance is significantly boosted via synthesized consistent views.

## 5 LIFT CONSISTENT MULTI-VIEWS TO 3D

**Native 3D Reconstruction**   Due to the inconsistent predictions by Zero123, there are numerous distortions in native 3D reconstruction like NeRF (Mildenhall et al., 2021) or NeuS (Wang et al., 2021). With Neus as an example in Figure 7(a), it shows that compared with Zero123, both the geometry and texture of constructed 3D models are greatly improved with the synthesized consistent views of Consistent123.

**Image-to-3D Generation**   We also show the 3D lifting results with recent SOTA image-to-3D methods, including DreamFusion (Poole et al., 2023) and One-2-3-45 (Liu et al., 2023a). It shows a great improvement in image-to-3D generation quality with Consistent123. In Figure 7(b) and 7(c), Zero123-based generation results (first row) encounter failure especially on the unseen side, due to the inconsistent synthesized views. With Consistent123 as the view synthesis backbone, the objects are successfully generated with relatively good quality (second row). Furthermore, we believe that with the boosting consistency, there still exists a large exploring space to improve the lifting quality.

## 6 CONCLUSION

In this work, we have presented Consistent123, an image-to-3D model to synthesize consistent multiple views. In the proposed model, synthesized views are aligned with cross-view attention and shared self-attention. We also designed two sampling strategies to support sampling arbitrary numbers of views and improve the view quality and consistency. Furthermore, we show that Consistent123 can serve as a new foundation model in varying downstream tasks like native 3D reconstruction and image-to-3D generation, which significantly boosts the quality of 3D lifting.

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

# A APPENDIX

## A.1 MORE QUALITATIVE RESULTS

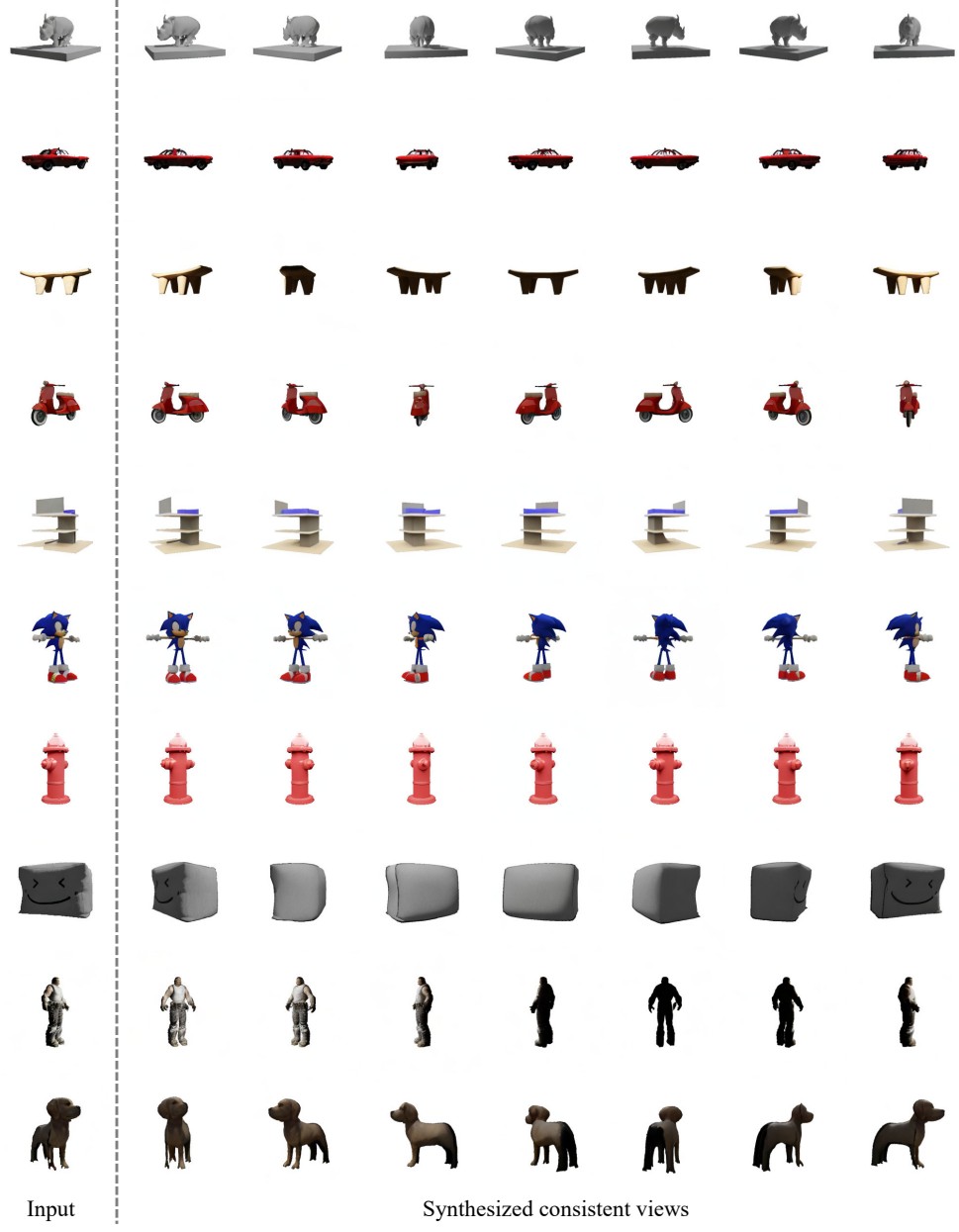

Input           Synthesized consistent views

Figure 8: **More qualitative results on Objaverse.**

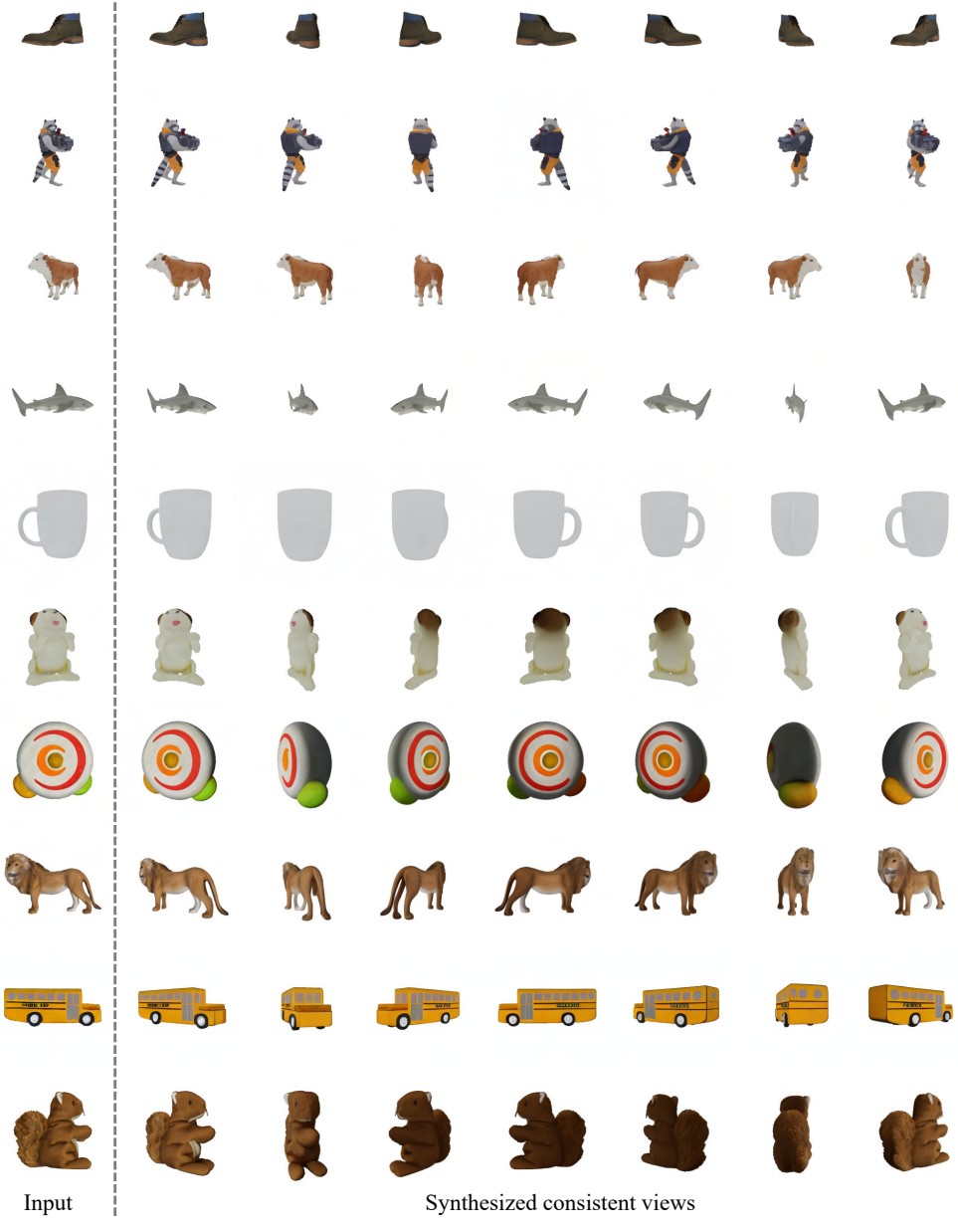

Input          Synthesized consistent views

Figure 9: **More qualitative results on GSO.**

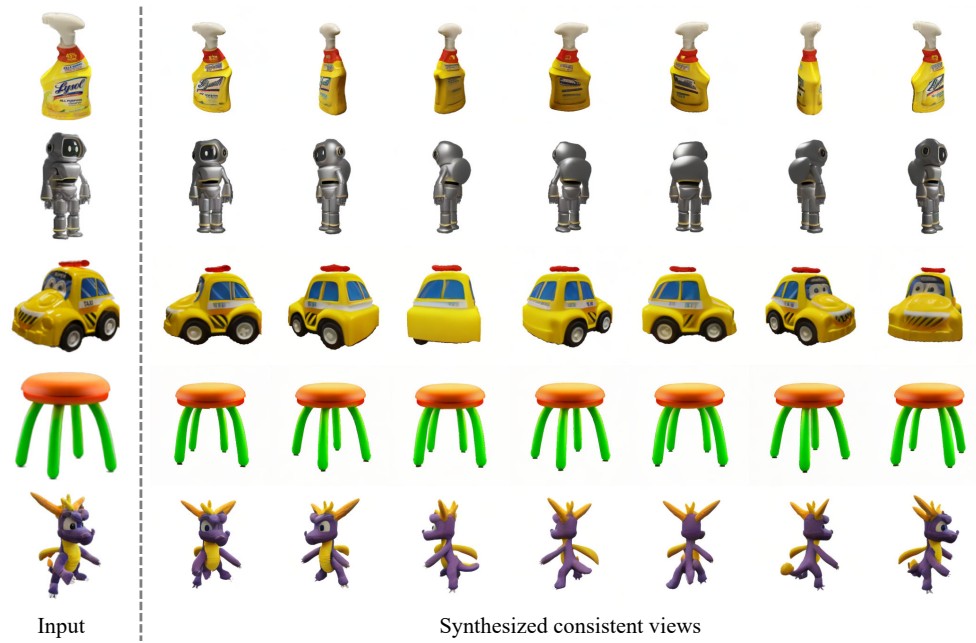

Input            Synthesized consistent views

Figure 10: **Qualitative results for images in the wild.**

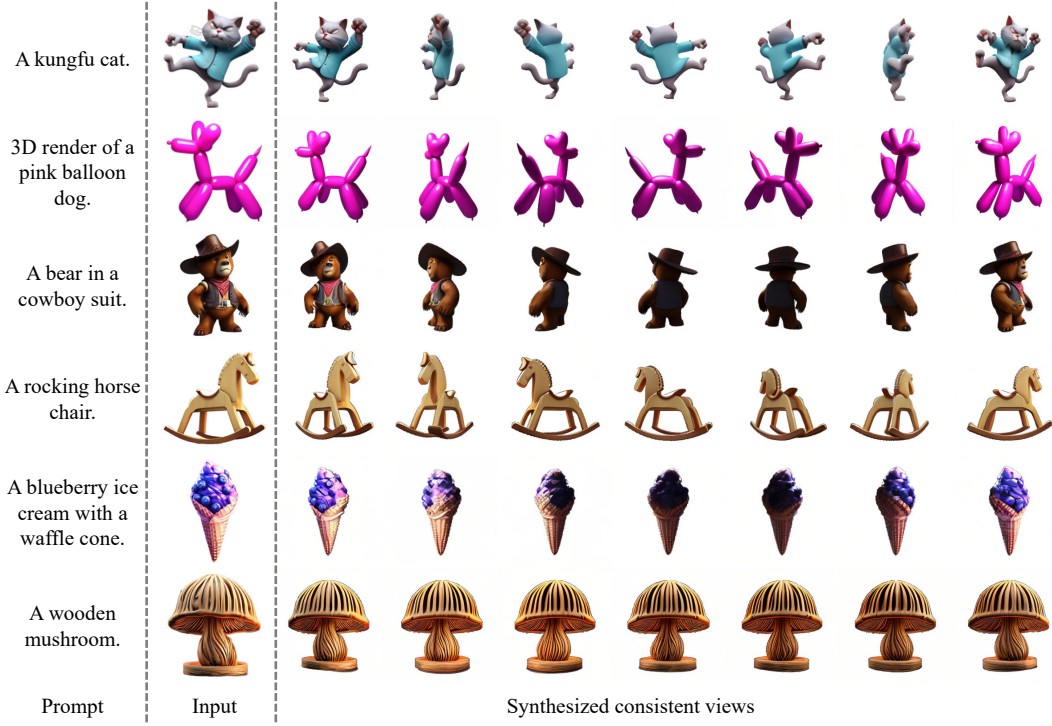

Prompt      Input          Synthesized consistent views

Figure 11: **Qualitative results for images generated by text-to-image models.**

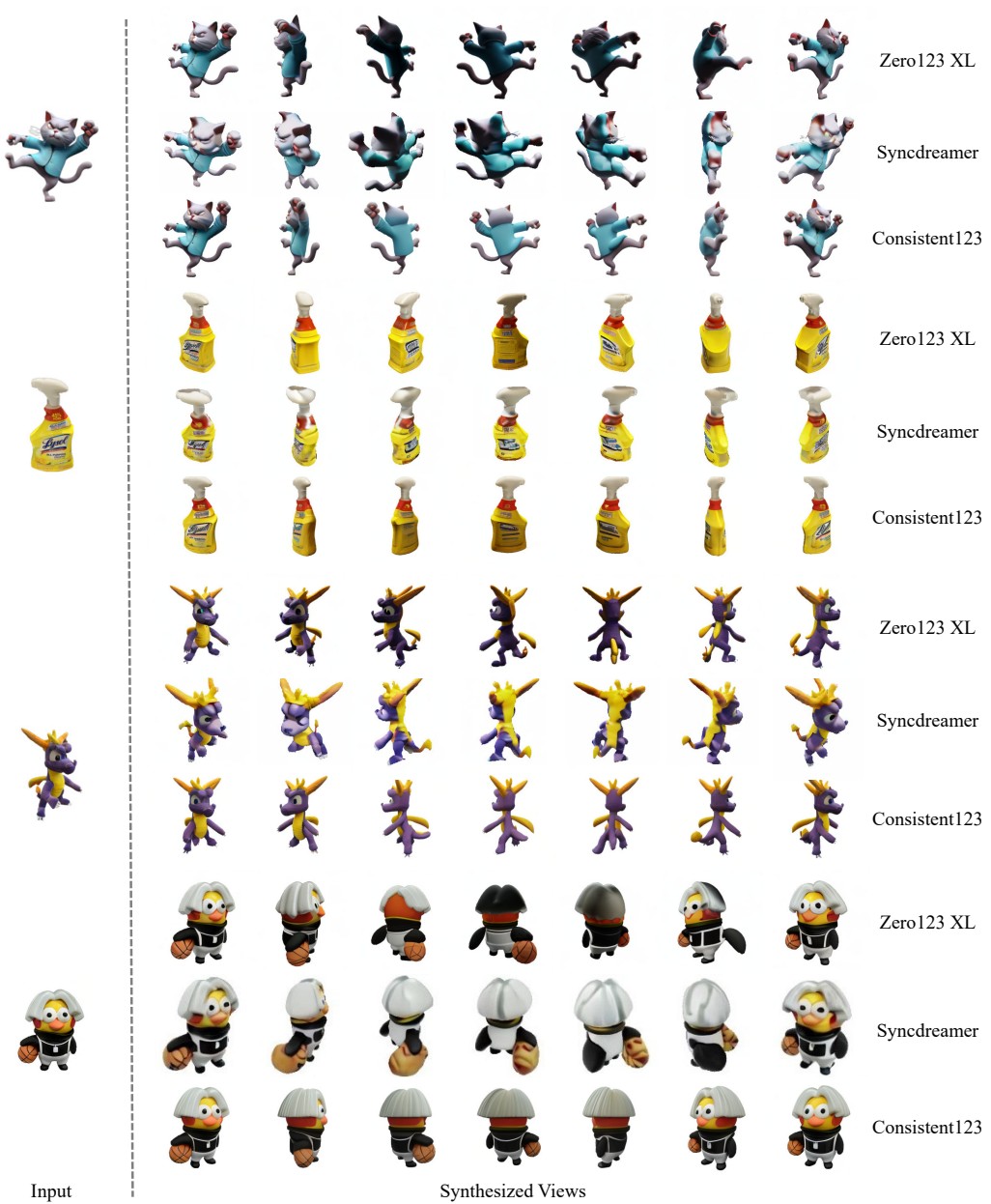

Figure 12: **Qaulitative comparison with the very recent baselines (Zero123 XL, Syncdreamer).** It shows that the proposed model can synthesize high-quality novel views with good consistency.

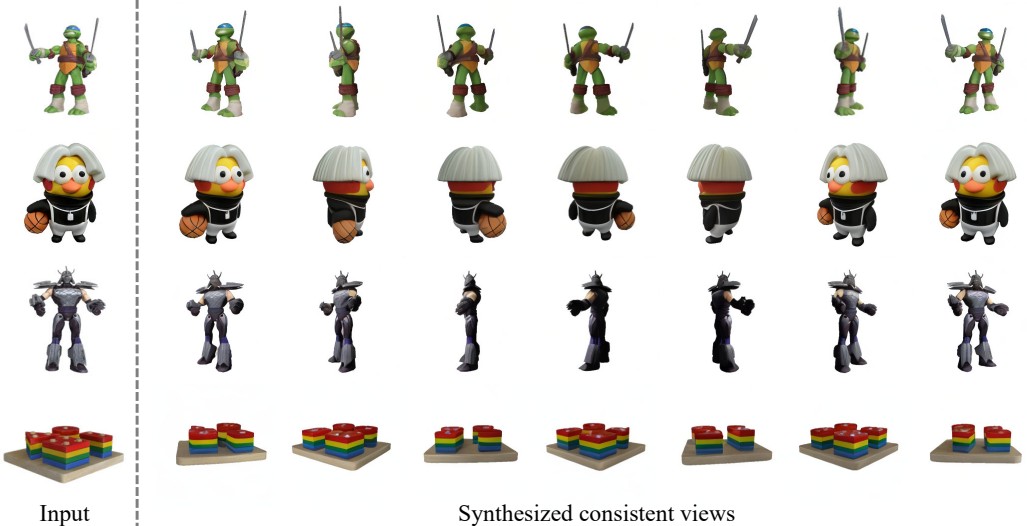

Input                        Synthesized consistent views

Figure 13: **Qualitative results for more sophisticated input images (e.g., sophisticated texture or spatial relation among objects).** It shows that the proposed method can be well generalized to more complicated input images.

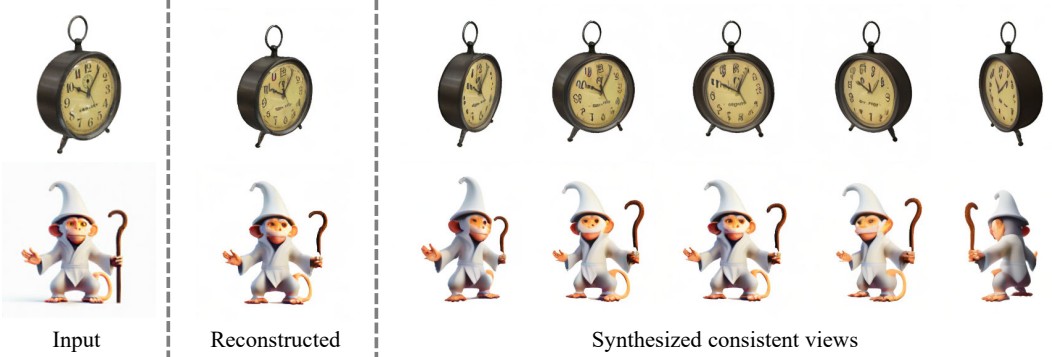

Input          Reconstructed                Synthesized consistent views

Figure 14: **Failure cases.** The proposed method fails to reconstruct some complicated patterns of the input images (e.g., text, thin objects).

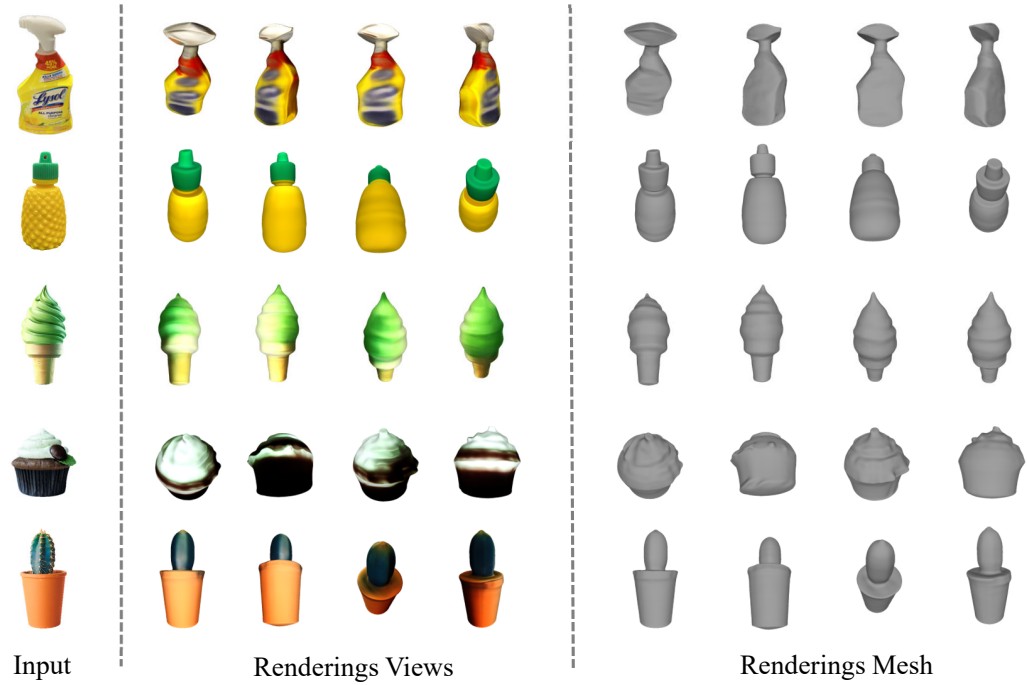

Input          Renderings Views          Renderings Mesh

Figure 15: **More qualitative results for 3D reconstruction.**

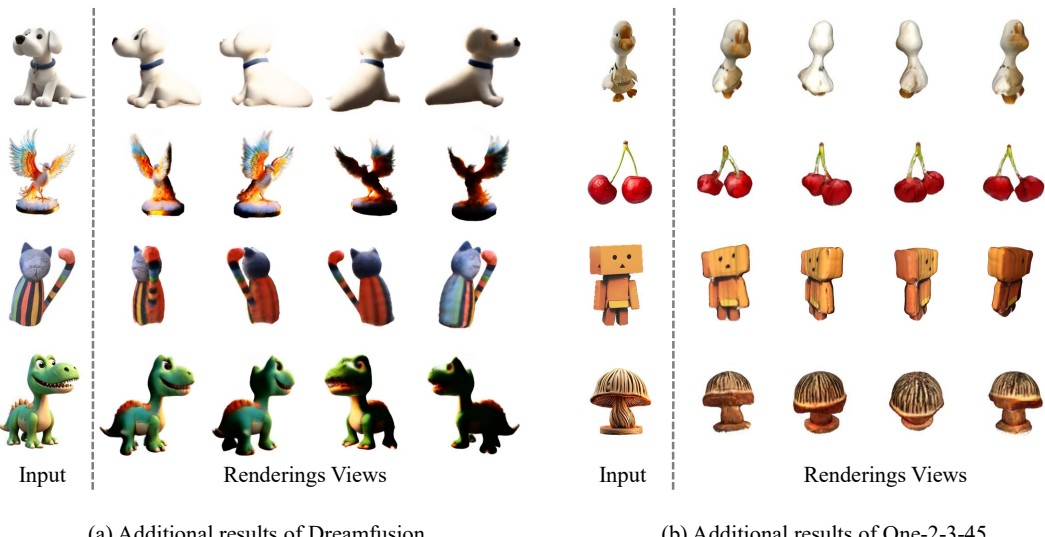

Input          Renderings Views          Input          Renderings Views

(a) Additional results of Dreamfusion          (b) Additional results of One-2-3-45

Figure 16: **More qualitative results for image-to-3D generation.**

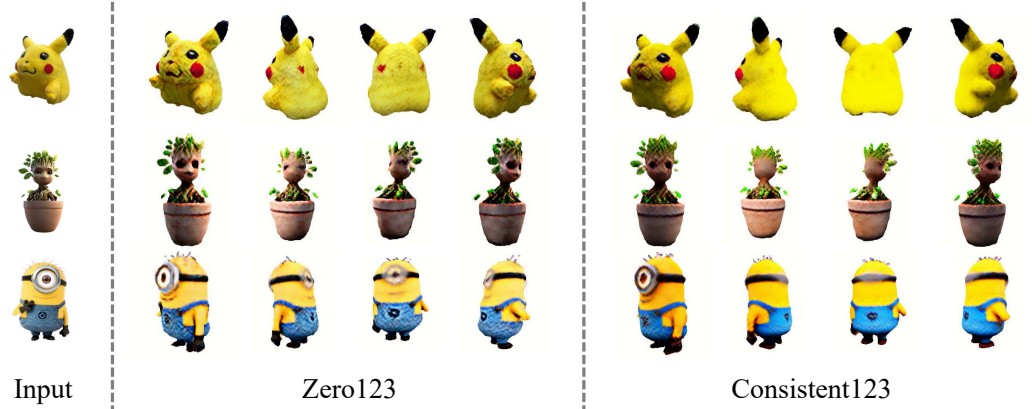

Input        Zero123        Consistent123

Figure 17: **Qualitative results for Magic123.** The proposed method greatly alleviates the multi-face Janus problem with the boosted view consistency.

## A.2 EVALUATION WITH DIFFERENT NUMBERS OF SAMPLING VIEWS

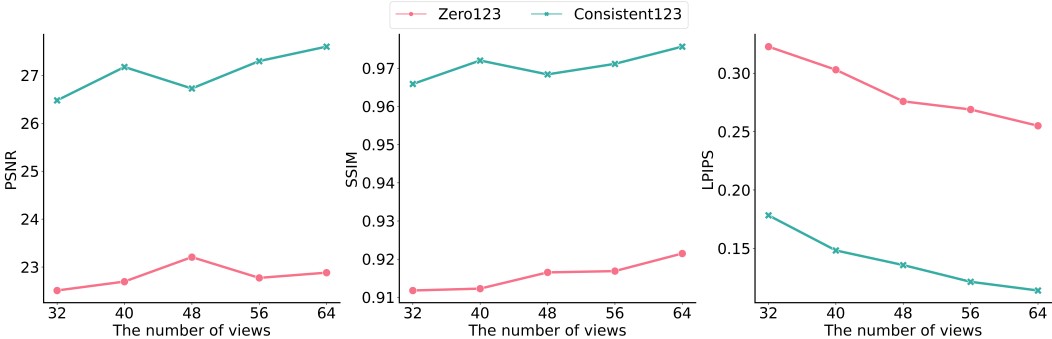

Figure 18: **The consistency score with different numbers of sampling views on GSO.** For the varying evaluation number of views, Consistent123 outperforms Zero123 by a large margin.

The consistency score in previous experiments was calculated with 64 sampling views. In Figure 18, we also show the consistency score on GSO measured in other numbers of sampling views. It shows that with different numbers of sampling views, our model consistently outperforms Zero123 by a large margin.

## A.3 IMPLEMENTATION DETAILS FOR IMAGE-TO-3D GENERATION

In Section 5, we show the boosted 3D lifting quality by replacing Zero123 with Consistent123. For image-to-3D generation methods, we make some necessary modifications to better utilize the power of Consistent123.

**DreamFusion.** DreamFusion (Poole et al., 2023) has been adapted to achieve image-to-3D generation. For each training step, vanilla DreamFusion uniformly picks the noise scale ranging from 0.02 to 0.98. In our early experiments, we found that such a noisy scale schedule would result in texture loss on the unseen side of objects. To solve this problem, we enlarged the noise scale at the early training stage. Specifically, the lower bound of the noise scale is initially set near the upper bound. As the training steps increase, the lower bound gradually decreases until approaches 0.02. This helps to find a good texture initialization for Consistent123 to improve the synthesis quality.

**One-2-3-45.** Vanilla One-2-3-45 (Liu et al., 2023a) synthesizes 8 views in the first stage and then predicts 4 nearby views for each of them in the second stage. This strategy prevents large pose

transformations of Zero123, thus alleviating the inconsistency to some extent, However, Consistent123 can directly synthesize all these views at once without losing view consistency and quality. So we directly generate all required views concurrently in one stage with Consistent123, and both the quality and speed are further improved.

## A.4    DIFFERENT TRAINING RENDERINGS OF OBJAVERSE

| Dataset | Objaverse Testset | | | GSO | | | RTMV | | |
|---|---|---|---|---|---|---|---|---|---|
| Rendering Pose | PSNR↑ | SSIM↑ | LPIPS↓ | PSNR↑ | SSIM↑ | LPIPS↓ | PSNR↑ | SSIM↑ | LPIPS↓ |
| Spherical Sampling | 24.18 | 0.95 | 0.17 | 25.82 | 0.96 | 0.15 | 17.84 | 0.84 | 0.28 |
| Circular Perturbation | **24.98** | **0.96** | **0.14** | **27.98** | **0.98** | **0.11** | **18.76** | **0.85** | **0.25** |

Table 5: **Consistency score of Consistent123 trained with different Objaverse renderings.** Training with circular perturbation renderings improves the view consistency compared with spherical sampling (training renderings of Zero123).

| Dataset | Objaverse Testset | | | GSO | | | RTMV | | |
|---|---|---|---|---|---|---|---|---|---|
| Rendering Pose | PSNR↑ | SSIM↑ | LPIPS↓ | PSNR↑ | SSIM↑ | LPIPS↓ | PSNR↑ | SSIM↑ | LPIPS↓ |
| Spherical Sampling | **16.71** | **0.87** | **0.14** | **18.27** | 0.87 | 0.13 | 9.71 | 0.66 | 0.34 |
| Circular Perturbation | 16.46 | 0.86 | **0.14** | 18.22 | 0.87 | 0.13 | 9.87 | 0.66 | 0.34 |

Table 6: **Ground truth comparison of Consistent123 trained with different Objaverse renderings.** The model trained with circular perturbation achieves similar performance compared with spherical sampling (training renderings of Zero123).

In Zero123 (Liu et al., 2023b), the random spherical renderings of Objaverse are used to train models. In this work, we render another version of Objaverse, whose poses are circularly sampled with perturbation as described in Section 4.1. As shown in Table 5 and 6, Consistent123 trained with circular perturbation renderings are more view-consistent while maintaining the similarity with the ground truth. Therefore, we select the circular perturbation renderings of Objaverse as our final training set to further improve the view consistency.

## A.5    MORE QUANTITATIVE COMPARISON WITH BASELINES.

**Comparison on novel view synthesis.**    To further show the effectiveness of Consistent123, we also conduct comparisons with the very recent baselines Zero123 XL and Syncdreamer. Note that Syncdreamer is constrained at fixed poses, thus we only compare with it qualitatively in Figure 12. In the following Table 7, we evaluate the consistency score on GSO dataset, which shows that Consistent123 also outperforms Zero123 XL by a large margin.

| Method | PSNR | SSIM | LPIPS |
|---|---|---|---|
| Zero123 | 22.88 | 0.92 | 0.25 |
| Zero123 XL | 24.13 | 0.94 | 0.21 |
| Consistent123 | **27.98** | **0.98** | **0.11** |

Table 7: **Quantitative comparison with the stronger baseline Zero123 XL.**

**Comparison on 3D reconstruction.**    Following the settings of Zero123, we calculate the Chamfer Distance and Volume IoU of image-to-3D reconstruction based on SDS loss in Table 8. Note that these metrics only evaluate the geometry quality, ignoring the texture improvement of Consistent123

| Method | Chamfer Distance↓ | Volume IoU↑ |
|---|---|---|
| Zero123 | 0.0880 | 0.4719 |
| Consistent123 | **0.0843** | **0.4818** |

Table 8: **Quantitative comparison with Zero123 on the dreamfusion-based 3D reconstruction.**

(which is more significant compared with geometry improvement). Please refer to the updated video supplementary (dreamfusion.mp4) for more qualitative comparisons.

### A.6 MORE ABLATION STUDY

**Constrained shared self-attention.** Intuitively, with a larger pose difference between the input view and synthesized view, the shared self-attention would become less activated. For these poses, the non-shared self-attention (i.e., U-Net Encoder part) and cross-view attention layers can serve as complementary and the performance can be maintained. For a better understanding of how shared self-attention works, we conducted an ablation study. For the poses whose azimuth angle with input view is larger than 90°, we set the shared self-attention to vanilla self-attention by force. We evaluate the consistency score on GSO in Table 9. The table shows that the constrained version of shared self-attention achieves similar performance as the vanilla self-attention. This indicates that the performance will degrade when disabling shared self-attention when a large pose difference exists, which makes the model harder to learn when attending keys and values from different views (half of the views attend to the input view, half attend to themselves).

| Method | PSNR | SSIM | LPIPS |
|---|---|---|---|
| Vanilla Self-attention | 24.64 | 0.94 | 0.17 |
| Constrained Shared Self-attention | 24.67 | 0.94 | 0.17 |
| Shared Self-attention | **27.98** | **0.98** | **0.11** |

Table 9: **Quantitative results on constrained shared self-attention.**

**The number of evaluation objects on Objaverse.** The evaluation of the consistency score is quite time-consuming (constructing a Nerf per object), preventing us from using a large number of evaluation objects. Moreover, our evaluation settings mainly borrow from Zero123, which only uses 20 objects in GSO (as well as most concurrent works). To demonstrate more convincing results, we extend the number of evaluation objects to 1k in Table 10, which shows that the proposed model consistently outperforms baselines by a large margin.

| Method | PSNR(100) | SSIM(100) | LPIPS(100) | PSNR (1000) | SSIM (1000) | LPIPS (1000) |
|---|---|---|---|---|---|---|
| Zero123 | 21.72 | 0.92 | 0.23 | 20.90 | 0.89 | 0.25 |
| Zero123 + SC | 22.09 | 0.92 | 0.21 | 20.78 | 0.89 | 0.24 |
| Consistent123 | 24.98 | 0.96 | 0.14 | 23.89 | 0.94 | 0.16 |

Table 10: **Quantitative results of consistency score for more Objaverse evaluation objects.**

**Trained from scratch.** For a fair comparison, we train Consistent123 from scratch on the same renderings as Zero123. We show the consistency score evaluated on GSO in Table 11. In our experiments, we found that it is challenging to simultaneously maintain the synthesized view quality and consistency. Due to the limited scale of the existing 3D dataset, the model is easy to be trapped in overfitting when directly trained from scratch. To reduce the overfitting, it is reasonable to enable the model with novel view synthesis ability by training like Zero123, then improve the view consistency

as Consistent123. Moreover, due to the slow training process (over 7 days for zero123), there is no further time to explore a more effective optimization strategy. A better training strategy (e.g., Efficient 3DiM) may make it possible to train Consistent123 from scratch.

| Method | PSNR | SSIM | LPIPS |
|---|---|---|---|
| Zero123 | 22.88 | 0.92 | 0.25 |
| Consistent123 (finetuned) | 25.82 | 0.96 | 0.15 |
| Consistent123 (from scratch) | 23.59 | 0.92 | 0.17 |

Table 11: **Quantitative results of Consistent123 trained from scratch**

**Trained with a dynamic number of views.** We also conducted the ablation study to use a random number of views during training. We implement it by selecting a random number of views dynamically for each batch, which ranges from 8 views to 16 views (since we only render 18 views for each object). We evaluate the consistency score on GSO as shown in Table 12. It shows that training at a fixed number of views is already effective, and training at a random view may degrade the performance. This is mainly because as the number of training views increases, the difference among views is smaller. It becomes easier for cross-view attention to predict the novel views, thus harming the robustness of the learned model and may lead to the overfitting issue.

| Method | PSNR | SSIM | LPIPS |
|---|---|---|---|
| Zero123 | 22.88 | 0.92 | 0.25 |
| Zero123 + SC | 22.30 | 0.93 | 0.21 |
| Consistent123 (8 views) | 27.98 | 0.98 | 0.11 |
| Consistent123 (8 16 views) | 26.25 | 0.96 | 0.14 |

Table 12: **Quantitative results of Consistent123 trained with dynamic number of views.**

