# OpenReview forum: "Consistent123: Improve Consistency for One Image to 3D Object Synthesis"
_ICLR.cc/2024/Conference — Submitted to ICLR 2024_

### Official Review · Reviewer_xWyo · 2023-10-29

**Soundness:** 3 good
**Presentation:** 2 fair
**Contribution:** 2 fair
**Rating:** 6
**Confidence:** 4

**Summary:**

Consistent123 is an improved version of Zero123 by optimizing using extra cross-attention consistency with a progressive classifier-free guidance strategy. This cross-attention training also enables generating an arbitrary number of views during inference.

**Strengths:**

- A clear improvement in novel view synthesis. The video shared in the supplementary file clears shows the strength of Consistent123 over Zero123.
- The cross-attention mechanism is simple yet effective.

**Weaknesses:**

1. Smoothness in Results. I noticed that the Consistent123 approach generally produces smoother results and seems to miss out on some finer details compared to Zero123. This observation is particularly evident in the first row of Fig. 4, and in the hat geometry depicted in Fig. 7: both in sections (a) bottom and (b) up. Have the authors identified potential strategies or modifications to address this shortcoming?

2. Concerns regarding arbitrary-length sampling. The methodology adopted uses a fixed number of views (8 views) during training. I'm concerned that this fixed view might adversely affect performance when dealing with arbitrary-length sampling at inference. This concern arises from a potential mismatch between training and test distributions. Any reason why using fixed number views during training? Is it only for simple implementation and faster training? An ablation study showcasing the performance with a random number of views as well during training would provide valuable insights and address this concern.

3. Ablation on Zero123 pretraining. Could you present results when Consistent123 is trained from scratch without Zero123 pretraining?

**Questions:**

1. Presence of Ground Truth for clarity. I would recommend including the Ground Truth in Figs 1, 4, 5, and 6. Having a point of reference would greatly enhance the clarity and allow for a more informed evaluation of the results.

2. Visualization of cross attention during training. The manuscript currently lacks visualizations for cross-attention dynamics during training. It would be beneficial for readers to see how these cross-attention maps evolve and converge throughout the training process.

3. It makes the paper stronger if you can show better 3D reconstruction results. For example, you can use your Consistent123 inside RealFusion [1] and Magic123 [2] to show state-of-the-art image-to-3D results.

Other minor suggestions:
1. suggest to add more views in Fig 1 since there are empty space. You can also point out in the figure where Zero123 fails and you success to catch the audiences’ attention quickly.
2. Show back views in Fig 7 (a) bottom and (b) up.
3. Better to add cross attention between views in Fig.2 (a) as well, like a few red lines across views at the denoised views.

[1] Melas-Kyriazi, Luke, Iro Laina, Christian Rupprecht, and Andrea Vedaldi. "Realfusion: 360deg reconstruction of any object from a single image." In Proceedings of the IEEE/CVF Conference on Computer Vision and Pattern Recognition, pp. 8446-8455. 2023.
[2] Qian, Guocheng, Jinjie Mai, Abdullah Hamdi, Jian Ren, Aliaksandr Siarohin, Bing Li, Hsin-Ying Lee et al. "Magic123: One image to high-quality 3d object generation using both 2d and 3d diffusion priors." arXiv preprint arXiv:2306.17843 (2023).

---

> ### Author Response · Authors · 2023-11-21
> **Reply to Reviewer xWyo**
>
> > Consistent123 may produce smooth texture in synthesized views for some cases. Have the authors identified potential strategies or modifications to address this shortcoming?
>
> It is an important but challenging task for NVS methods to predict novel views with finer details. To further solve the smooth texture problem, the following techniques may help:
>
> - Lower the PCFG guidance scale and increase inference steps. For all cases in the paper, we use the fixed schedule for progressive classifier-free guidance without per-case tuning to perform fair comparisons.  For some specific cases as mentioned above, the starting and ending points of progressive classifier-free guidance can be tuned (lower for better texture) to alleviate the smoothness problem. Moreover, using larger DDIM sampling steps (e.g., 200) also helps to improve the synthesized texture.
> - Improve the quality of training data. As mentioned in other concurrent works (e.g., Instant3D), there is a lot of low-quality training data in Objaverse for novel view synthesis, which may result in the smoothness of the synthesized texture. A careful dataset pruning strategy may further alleviate this problem.
>
> >  What is the performance with a random number of views as well during training?
>
> It is interesting to use a random number of views during training. We implement it by selecting a random number of views dynamically for each batch, which ranges from 8 views to 16 views (since we only render 18 views for each object). We evaluate the consistency score on GSO as shown in the following table. It shows that training at a fixed number of views is already effective, and training at a random view may degrade the performance. This is mainly because as the number of training views increases, the difference among views is smaller. It becomes easier for cross-view attention to predict the novel views, thus harming the robustness of the learned model and may lead to the overfitting issue.
>
> |           Method           | PSNR  | SSIM | LPIPS |
> | :------------------------: | :---: | :--: | :---: |
> |          Zero123           | 22.88 | 0.92 | 0.25  |
> |        Zero123 + SC        | 22.30 | 0.93 | 0.21  |
> |  Consistent123 (8 views)   | 27.98 | 0.98 | 0.11  |
> | Consistent123 (8~16 views) | 26.25 | 0.96 | 0.14  |
>
> > Can Consistent123 be trained from scratch?
>
> For a fair comparison, we train Consistent123 from scratch on the same renderings as Zero123. We show the consistency score evaluated on GSO in the following table:
>
> |              Method               | PSNR  | SSIM | LPIPS |
> | :-------------------------------: | :---: | :--: | :---: |
> |   Zero123 (from scratch, 105k)    | 22.88 | 0.92 | 0.25  |
> |   Consistent123 (finetune, 10k)   | 25.82 | 0.96 | 0.15  |
> | Consistent123 (from scratch, 20k) | 23.59 | 0.92 | 0.17  |
>
> In our experiments, we found that it is challenging to simultaneously maintain the synthesized view quality and consistency. Due to the limited scale of the existing 3D dataset, the model is easy to be trapped in overfitting when directly trained from scratch. To reduce the overfitting, it is reasonable to enable the model with novel view synthesis ability by training like Zero123, then improve the view consistency as Consistent123. Moreover, due to the slow training process (over 7 days for zero123), there is no further time to explore a more effective optimization strategy. A better training strategy (e.g., Efficient 3DiM) may make it possible to train Consistent123 from scratch.
>
> > It makes the paper stronger if you can show better 3D reconstruction results (e.g. use Magic123).
>
> Better 3D reconstruction results using Magic123 are shown in the video supplementary (magic123.mp4) and Figure 17.
>
> > Minor suggestions mentioned in the review.
>
> We thank you for all the kind suggestions and will consider them in the revised paper.

---

### Official Review · Reviewer_n2pA · 2023-10-31

**Soundness:** 3 good
**Presentation:** 3 good
**Contribution:** 2 fair
**Rating:** 5
**Confidence:** 4

**Summary:**

In this paper, the authors aim to improve the view consistency for the novel view synthesis method based on image-to-image translation (i.e., Zero123). Specifically, They incorporate Zero123 with shared self-attention layers and additional cross-view attention layers. In addition, they propose a progressive classifier-free guidance strategy to balance the texture and geometry during the denoising process. Experimental results show that the proposed Consistent123 achieves better view consistency on multiple benchmarks compared to Zero123. The authors also demonstrate the potential of Consistent123 on various downstream tasks, such as 3D Reconstruction and image-to-3D generation.

**Strengths:**

1. The proposed method allows flexible view numbers compared to concurrent work MVDream. Experiments show that using arbitrary-length sampling with more view numbers could boost view consistency, indicating the effectiveness of the proposed method.
2. The proposed method is intuitive yet effective. By adding additional attention mechanisms, the authors improve the view consistency of Zero123.
3. The proposed progressive classifier-free guidance is interesting and alleviates the trade-off between geometry and texture.

**Weaknesses:**

1. The attention mechanisms are totally borrowed from previous work, such as shared self-attention from Cao et al. and cross-attention from video diffusion models.
2. For the shared self-attention layers, when the views are totally orthogonal, how will this shared self-attention act? Can this self-attention find correct correspondence? For example, in Figure 5 (right), when there is no shared self-attention, the resulting first view seems much more interesting. It would be better to have self-attention visualization in this case.
3. Considering Objaverse has 800K+ 3D models, the authors only picked up 100 objects for Table 1, which seems far from enough.
4. The proposed Consistent123 loads pretrained weight from Zero123 and fixes these weights. It would be fair to also have a version training from scratch.
5. Will the compromise solution introduce view inconsistency? It looks like there is no connection between the sampled views and the next round views.
6. The results on 3D reconstruction seem poor, where the results from Neus are blurry and low-quality.

**Questions:**

1. For the shared self-attention layers, is there any positional embedding? If not, does introducing a camera pose-aware positional embedding help? Do you have any insight on this?
2. Since the current conditions on R and T are still geometry-free, I am worried that the proposed method's upper bound is limited.

---

> ### Author Response · Authors · 2023-11-21
> **Reply to Reviewer n2pA**
>
> > The attention mechanisms in Consistent123 are borrowed from the video diffusion model and Cao et al.
>
> Although the high-level idea of attention mechanisms is inspired by video generation, they still remain unexplored in 3D generation and can not be directly applied for novel view synthesis. We make several key modifications to adapt these components for better view consistency.
>
> - Shared self-attention is an inference-only strategy in Cao et al., but we adapt it for both training and inference in Consistent123 to further enhance the alignment between novel views and input views.
>
> - To attend to the input view latent, we fix the first pose of the view sequence to be the same as the input image, thus avoiding additional inverse operations and making the training process more efficient.
>
> - We apply shared self-attention in all the diffusion timesteps, which only part of timesteps are used in Cao et al. We also conduct an ablation study in Table 3(b) to decide the position to apply for shared self-attention.
>
> - Finally, other important components of Consistent123 also significantly contribute to the view consistency, including the flexible number and pose at inference (while the frame length in video generation is typically fixed) and progressive classifier-free guidance.
>
> > How will the shared self-attention act if there exists a large pose difference between the input view and the synthesized view?
>
> Intuitively, with a larger pose difference between the input view and synthesized view, the shared self-attention would become less activated. For these poses, the non-shared self-attention (i.e., U-Net Encoder part) and cross-view attention layers can serve as complementary and the performance can be maintained.
>
> For a better understanding of how shared self-attention works, we conducted an ablation study. For the poses whose azimuth angle with input view is larger than 90°, we set the shared self-attention to vanilla self-attention by force. We evaluate the consistency score on GSO in the following table. The table shows that the constrained version of shared self-attention achieves similar performance as the vanilla self-attention. This indicates that the performance will degrade when disabling shared self-attention when a large pose difference exists, which makes the model harder to learn when attending keys and values from different views (half of views attend to input view, half attend to themselves).
>
> |              Method               | PSNR  | SSIM | LPIPS |
> | :-------------------------------: | :---: | :--: | :---: |
> |      Vanilla Self-attention       | 24.64 | 0.94 | 0.17  |
> |       Shared Self-attention       | 27.98 | 0.98 | 0.11  |
> | Constrained Shared Self-attention | 24.67 | 0.94 | 0.17  |
>
> > The authors only picked up 100 objects for Table 1, which seems far from enough.
>
> We also agree that more evaluation objects help to reduce the randomness of quantitative results. However, the evaluation of the consistency score is quite time-consuming (constructing a Nerf per object), preventing us from using a large number of evaluation objects. Moreover, our evaluation settings is mainly borrow from Zero123, which only use 20 objects in GSO (as well as most concurrent works).
>
> To demonstrate more convincing results, we extend the number of evaluation objects to 1k in the following table, which shows that the proposed model consistently outperforms baselines by a large margin.
>
> |    Method     | PSNR (100) | SSIM (100) | LPIPS (100) | PSNR (1000) | SSIM (1000) | LPIPS (1000) |
> | :-----------: | :--------: | :--------: | :---------: | :---------: | :---------: | :----------: |
> |    Zero123    |   21.72    |    0.92    |    0.23     |    20.90    |    0.89     |     0.25     |
> | Zero123 + SC  |   22.09    |    0.92    |    0.21     |    20.78    |    0.89     |     0.24     |
> | Consistent123 |   24.98    |    0.96    |    0.14     |    23.89    |    0.94     |     0.16     |

---

> > ### Author Response · Authors · 2023-11-21
> > **Reply to Reviewer n2pA**
> >
> > > Can Consistent123 be trained from scratch?
> >
> > For a fair comparison, we train Consistent123 from scratch on the same renderings as Zero123. We show the consistency score evaluated on GSO in the following table:
> >
> > |            Method            | PSNR  | SSIM | LPIPS |
> > | :--------------------------: | :---: | :--: | :---: |
> > |           Zero123            | 22.88 | 0.92 | 0.25  |
> > |  Consistent123 (finetuned)   | 25.82 | 0.96 | 0.15  |
> > | Consistent123 (from scratch) | 23.59 | 0.92 | 0.17  |
> >
> > In our experiments, we found that it is challenging to simultaneously maintain the synthesized view quality and consistency. Due to the limited scale of the existing 3D dataset, the model is easy to be trapped in overfitting when directly trained from scratch. To reduce the overfitting, it is reasonable to enable the model with novel view synthesis ability by training like Zero123, then improve the view consistency as Consistent123. Moreover, due to the slow training process (over 7 days for zero123), there is no further time to explore a more effective optimization strategy. A better training strategy (e.g., Efficient 3DiM) may make it possible to train Consistent123 from scratch.
> >
> > > Will the compromise solution introduce view inconsistency?
> >
> > - First, Consistent123 can synthesize highly consistent multi-views when the number of synthesized views is large. Although there is no direct connection between sampled views and next-round views, the pose difference between the next-round view would be very small, thus the inconsistency can be ignorable.
> > - Second, our models can generate more than 100 views simultaneously, which to our best knowledge is one of the largest among all concurrent methods and totally enough for common cases.
> >
> > > The results on 3D reconstruction seem poor, where the results from Neus are blurry and low-quality.
> >
> > - 3D reconstruction with direct pixel supervision (e.g., Neus) is quite challenging since it requires exact pixel-level consistency among the generated multi-views. For our baseline Zero123, the reconstruction results of Neus are much worse. With the boosted consistency of the proposed method, the 3D reconstruction quality would be greatly improved. For a qualitative comparison with the reconstruction result of Zero123, please refer to the updated video supplementary.
> > - The quality of these tasks downstream tasks is highly dependent on their implementation and hyper-parameter settings. To show more convincing reconstruction results, we apply Consistent123 to the more advanced reconstruction method Magic123, whose results are shown in the video supplementary and Figure 17.
> >
> > > For the shared self-attention layers, is there any positional embedding?
> >
> > No positional embedding is introduced in Consistent123 and it seems not quite necessary. Actually, pose-aware positional embeddings (R and T) have been introduced at cross-attention layers, and they would be used implicitly in shared self-attention and cross-view attention layers. It may be redundant to add pose embeddings again at shared self-attention and cross-view attention layers.
> >
> > > Geometry-free pose conditioning may limit the the upper bound of Consistent123.
> >
> > Although the view consistency may be possibly limited by the geometry-free pose conditioning, our method achieves the trade-off between view quality and consistency. Geometry-aware methods may suffer from the following issues:
> >
> > - The view quality may be relatively low compared with geometry-free methods (see the qualitative comparison with the recent geometry-aware method syncdreamer in Figure 12).
> > - It is quite time-consuming (10x more than geometry-free methods) in both the training and inference process.
> > - The pose and number of views are often restricted in geometry-aware methods.
> >
> > Therefore, we adopt the same geometry-free conditioning as zero123 in a purely data-driven manner.

---

### Official Review · Reviewer_4aQC · 2023-10-31

**Soundness:** 3 good
**Presentation:** 3 good
**Contribution:** 3 good
**Rating:** 5
**Confidence:** 5

**Summary:**

The paper introduced a way that improves novel view synthesis model, e.g. zero123 by considering a multiview input in the diffusion model. While consider a shared self-attention machanism that all views  query the same key and value from the input view, which provides detailed spatial layout information for novel view synthesis.  In the method, It supports Arbitrary-length Sampling and adopted Progressive Classifier-free Guidance, yielding a further improvement of the synthesis.

The resulting novel views looks more consistent than baseline. And from the supplimentary material.

**Strengths:**

1. The multiview input to the diffusion is good in achieving geometric consistency, comparing against zero123 base model.

2. The design of progressive scheduler is interesting by jointly considering the benefit from large cfg vs small cfg, which leverage between texture and geometry.

3. The paper demonstrates through qualitative and quantitative experiments that Consistent123 significantly outperforms baselines, zero123 in particular, in view consistency, showcasing substantial improvement in various downstream tasks.

**Weaknesses:**

Novelty is clear, while there are several publications available with open-sourced papers. Such as magic123,  zero123 xl, sync-dreamer.   for synthesizing new views and do the 3D reconstruction using SDS or direct pixel loss based on NeuS. Wonder the author may compare the results with the opensourced recon-models.

From the experimental results after 3D reconstruction. It looks like a black biased back side are generated. Which in my perspective, they are no better than the pulished methods such as that has been implemented in threestudio [url: https://github.com/threestudio-project/threestudio] [which is available before the submission].  The autho may explain why the reconstructed results

**Questions:**

1. The generated views are still not fully consistent before the 3D model is reocnstructed, while the renderred image after reconstruction looks much worse.  is there any thoughts in further improve the consistency so the quality gap between diffused output and render-view can be minimized ?

2. How it generalizes towards more sophisticated images ?  Please also provide some faliure cases for a better understand of the limitations.

---

> ### Author Response · Authors · 2023-11-21
> **Reply to Reviewer 4aQC**
>
> > Can Consistent123 be compared with recent work magic123, zero123 xl, sync-dreamer?
>
> - Magic123 is an SDS-based method for image-to-3D based on Zero123, which can be regarded as a downstream application of Consistent123. Therefore, we show the experiment results of Magic123 based on Consistent123 rather than directly comparing with it. Please refer to the video supplementary (magic123.mp4) and Figure 17 for more details.
>
> - Both Zero123 XL and Syncdreamer are used for novel view synthesis, so we directly compare them in synthesized multi-views. Note that Syncdreamer is constrained at fixed poses, thus we only compare with it qualitatively. Please refer to supplementary Figure 12 for a qualitative comparison with these baselines.
>
> Besides the qualitative comparison in Figure 17 and Figure 12, we also evaluate the consistency score on GSO dataset for the quantitative comparison.
>
> |    Method     | PSNR  | SSIM | LPIPS |
> | :-----------: | :---: | :--: | :---: |
> |    Zero123    | 22.88 | 0.92 | 0.25  |
> |  Zero123 XL   | 24.13 | 0.94 | 0.21  |
> | Consistent123 | 27.98 | 0.98 | 0.11  |
>
> > Why the reconstructed results are no better than the published methods in threestudio?
>
> The reconstruction results of dreamfusion are highly dependent on hyper-parameter tuning. For a fair comparison, all the dreamfusion experiments shown in the paper follow the same hyper-parameter settings without further per-case tuning (and note that all these experiments are also conducted under the implementation of threestudio). As for some failure textures, we could use stronger optimization-based reconstruction methods (e.g., Magic123) to improve the quality. To further demonstrate the effectiveness of Consistent123, we show better 3D reconstruction results using Magic123 in the video supplementary (magic123.mp4) and Figure 17.
>
> > Are there any thoughts on further improving the consistency so the quality gap between diffused output and render-view can be minimized?
>
> - First, introducing geometry-aware conditioning may help. For instance, spatial volume or epipolar attention can be used to further enhance the consistency. However, such architecture may reduce the quality and flexibility of synthesized views (e.g., the number and pose of view must be fixed at inference).
> - Second, a better data filtering strategy may help. There are lot of low-quality data in our training dataset Objaverse as indicated by other concurrent methods (e.g., Instant3D). Training on the full dataset of Objaverse may result in oversmoothed textures and increase the difficulty of being consistent.
>
> > How it generalizes towards more sophisticated images?
>
> More sophisticated cases are shown in supplementary Figure 13, indicating that Consistent123 can be generalized to sophisticated images while preserving good consistency.
>
> > Please also provide some failure cases for a better understanding of the limitations.
>
> Failure cases are provided in supplementary Figure 14. The main problem falls on the latent diffusion decoder, which struggles to reconstruct some complicated pattern of input images (e.g., text, thin objects). A direct solution is to use a stronger latent decoder like the Dalle3 decoder (OpenAI). Using a cascaded diffusion model may be another solution.

---

### Official Review · Reviewer_bDyq · 2023-10-31

**Soundness:** 3 good
**Presentation:** 3 good
**Contribution:** 2 fair
**Rating:** 6
**Confidence:** 4

**Summary:**

This paper introduces a novel method to synthesize a set of images of any objects from novel view given a single image as input. One of many challenges in this task is to generate consistent images in terms of geometry and appearance. To this end, the authors propose to generate multiple images simultaneously and enable cross-attention between novel images at different viewpoints. To strike a balance between geometry and texture of generated images, they also propose a progressive Classifier Free Guidance (CFG) after observing that a larger CFG often leads to better geometry but poor texture and a smaller CFG causes an opposite result. Experiments demonstrate that the proposed method outperform a popular baseline,  Zero123, on image synthesize.

**Strengths:**

- Consistency in novel view synthesis is at the core of many image-to-3D task. The proposed method is able to outperform Zero123 by a large margin qualitatively and quantitatively.
- The observation of more generated views improving consistency Is useful to other image- or text-conditioned novel view synthesis works.
- Paper is generally easy to follow

**Weaknesses:**

- V_c should be after softmax in Eq. 5.
- The name "shared self-attention" is confusing to me. It in fact is a cross-attention from the novel views to the input view. Why is it called self-attention?
- Only qualitative results were presented for image-to-3D tasks.

**Questions:**

- How many views were used in the NeuS experiment?

- No texture on the spray bottle in Figure 7?

- Is the Super Mario a failure case since the object has flattened? Could it be related to the progressive CFG? Figure 3 seems to suggest that a large CFG could lead to flat objects.

---

> ### Author Response · Authors · 2023-11-21
> **Reply to Reviewer bDyq**
>
> > V_c should be after softmax in Eq. 5
>
> Thanks for your careful reading and pointing out the typo. We will fix it in the revised paper.
>
> > Why is the name "shared self-attention"? It seems a cross-attention from the novel views to the input view.
>
> The shared self-attention is similar to the cross-attention between the novel view and input view, but they have several key differences:
>
> - It reuses the pre-trained weight of vanilla self-attention layers in Zero123.
> - It is non-trainable but only changes the calculation of self-attention.
>
> Therefore, we name it “shared self-attention” to distinguish these differences.
>
> > Only qualitative results were presented for image-to-3D tasks.
>
> Following the settings of Zero123, we calculate the Chamfer Distance and Volume IoU of image-to-3D reconstruction based on SDS loss. Note that these metrics only evaluate the geometry quality, ignoring the texture improvement of Consistent123 (which is more significant compared with geometry improvement). Please refer to the updated video supplementary (dreamfusion.mp4) for more qualitative comparisons.
>
> |    Method     | Chamfer Distance ↓ | Volume IoU ↑ |
> | :-----------: | :----------------: | :----------: |
> |    Zero123    |       0.0880       |    0.4719    |
> | Consistent123 |     **0.0843**     |  **0.4818**  |
>
> > How many views were used in the NeuS experiment?
>
> 3D reconstruction methods like Nerf and Neus typically require a large number of views. Therefore, we use 64 views for all Neus experiments.
>
> > No texture on the spray bottle in Figure 7?
>
> In the spray bottle case, we deliberately remove the texture from the object to show the synthesized geometry of Neus, which more concentrates on geometry reconstruction. We also add the textured version of this case into the supplementary Figure 15.
>
> > Is the Super Mario a failure case since the object has flattened?
>
> Super Mario is not a failure case. As a Lego toy, it is “flat” by design (feel free to google for more Lego Mario images). This case also shows that the object geometry can be constructed more reasonably with improved consistency.

---

### Author Response · Authors · 2023-11-21
**Overall Reply**

We would like to thank all the reviewers for their constructive comments and valuable feedback.

According to the reviews, we have made the following modifications to our paper (marked as blue in revised paper):
- add Figure 12, 13, 14, 17
- add Table 7, 8, 9, 10, 11, 12
- add supplementary Section A.5, A.6
- update Figure 7, 15

---

### Author Response · Authors · 2023-11-22

Dear My Reviewer bDyq, 4aQC, n2pA, xWyo,

Thank you for your support and helpful comments. We've tried our best to address your concerns, and we hope our responses make sense to you. Importantly, we much value your comments and would be happy to discuss more. If you have any additional questions or open discussions, please don't be hesitant to leave more comments. We are always available at all time, to actively address any concerns or be prepared for more discussions.

Your opinions are rather important for us to improve the work!

Thank you!

Sincerely,

Authors

---

### Meta-Review · Area_Chair_wEg8 · 2023-12-10

**Metareview:**

The paper was reviewed by 4 domain experts. The scores put the paper at borderline, slightly leaning negative. Reviewers acknowledged that consistency is an important aspect in novel view synthesis. However, they also mentioned that the results are more smooth as also admitted by authors. The authors provided their response, addressing some of the comments. After the discussion period, the reviewers shared their concerns the generated results are not impressive, insufficient experimental results and again suggested that the paper is at the borderline. With these inputs, the AC decides to reject the paper as there are no sufficient grounds for acceptance.

**Justification For Why Not Higher Score:**

The paper didn't receive the necessary support from reviewers. The AC read comments and the responses and confirmed that paper is at the borderline. Even positive reviewers in their message to AC were not so positive and suggested that results are not impressive.

**Justification For Why Not Lower Score:**

N/A

---

### Decision · Program_Chairs · 2024-01-16

Reject